# Open-Source Implementations of the Reactive Asset Administration Shell: A Survey

**DOI:** 10.3390/s23115229

**Published:** 2023-05-31

**Authors:** Michael Jacoby, Michael Baumann, Tino Bischoff, Hans Mees, Jens Müller, Ljiljana Stojanovic, Friedrich Volz

**Affiliations:** Fraunhofer Institute of Optronics, System Technologies and Image Exploitation IOSB, 76131 Karlsruhe, Germany; michael.baumann@iosb.fraunhofer.de (M.B.); tino.bischoff@iosb.fraunhofer.de (T.B.); ljiljana.stojanovic@iosb.fraunhofer.de (L.S.); friedrich.volz@iosb.fraunhofer.de (F.V.)

**Keywords:** Industry 4.0, digital twin, asset administration shell, AAS, implementation, open source

## Abstract

The use of open-source software is crucial for the digitalization of manufacturing, including the implementation of Digital Twins as envisioned in Industry 4.0. This research paper provides a comprehensive comparison of free and open-source implementations of the reactive Asset Administration Shell (AAS) for creating Digital Twins. A structured search on GitHub and Google Scholar was conducted, leading to the selection of four implementations for detailed analysis. Objective evaluation criteria were defined, and a testing framework was created to test support for the most common AAS model elements and API calls. The results show that all implementations support at least a minimal set of required features while none implement the specification in all details, which highlights the challenges of implementing the AAS specification and the incompatibility between different implementations. This paper is therefore the first attempt at a comprehensive comparison of AAS implementations and identifies potential areas for improvement in future implementations. It also provides valuable insights for software developers and researchers in the field of AAS-based Digital Twins.

## 1. Introduction

In recent years, the Digital Twin (DT), as one of the cornerstone technologies of Industry 4.0 (I4.0), has continued on its path from theoretical concept into real-world application in the context of industrial production. At the same time, there is an emerging consensus that the development and use of free and open-source software is crucial for this digitalization of manufacturing [1]. In this paper, we therefore investigate existing free and open-source implementation of the Asset Administration Shell (AAS) specification which is a concrete adaptation of the generic DT concept focusing on industrial production.

The AAS specification developed by the Plattform Industrie 4.0, a network of companies, associations, trade unions, science, and politics in Germany, is non-proprietary, platform-independent, and royalty-free [2,3]. It defines three types of AAS as shown in Figure 1: Passive AAS (file-based serialization of the AAS), Reactive AAS (executable AAS that can be communicated with via APIs), and Proactive AAS (executable AAS that can actively communicate via I4.0 languages). The AAS specification is also structured into different parts: Part 1 defines the AAS metamodel and serialization formats, while Part 2 specifies the APIs for reactive AASs. A simplified version of the AAS metamodel is shown in Figure 2. The main element is the *AssetAdministrationShell* which is linked to an asset via the *AssetInformation*. It is made up of multiple *Submodel*s which itself are comprised of several *SubmodelElement*s. There are multiple different types of submodel elements of which *Property* and *Operation* are depicted as examples. Submodels can exist independently of an AAS as indicated by the aggregation relationship between them, whereas submodel elements can only exist in the context of a submodel.

Over recent years, awareness of the importance of free and open-source software has increased, e.g., the EU emphasizes the importance and promotes the use of open-source software to “transition to an inclusive, better digital environment that is ready for the realities of today’s global economy” [5]. As the insight that “open-source software is at the heart of how we run our lives” [6] emerges in society, this also reflects in the attitude towards open source in the industry. A recent study by Bitkom e.V. [7] states that “open source is a decisive factor in shaping digitalization successfully” and that 7 out of 10 companies (with 20+ employees) deliberately use open-source solutions with the primary motivation being cost savings and openness of the software. Besides the increasing relevance in companies and industry, open-source software is gaining a more and more important role in the development of open standards. “The understanding now seems to be that standardization and open source are coming closer together to be more effective in both developing and promoting open specifications, open standards and open source” [8]. This interactive approach between open standards and open-source software called agile standardization is currently being implemented in multiple (standards development) organizations such as IETF, OASIS, Eclipse, and the Linux Foundation [9,10,11,12]. An additional, presumably event stronger, argument why open-source DT implementations are essential is the fact that DTs are designed to establish interoperability by providing a standardized way to exchange information along the whole value chain which includes many companies. Today, multiple organizations are being formed or already exist to promote open standards and develop open-source implementations such as the Industrial Digital Twin Association (IDTA) [13] and Digital Twin Consortium (DTC) [14].

In this paper, we investigate existing free and open-source implementations of the reactive AAS resp. Part 2 of the specification entails support of Part 1 (more precisely, at least supporting the metamodel as well as JSON de-/serialization) and compares them based on the proposed evaluation criteria.

Our main contributions are two-fold. First, we propose a holistic framework for testing and evaluating different AAS implementations. The framework comprises a set of relevant evaluation criteria, an AAS model covering all aspects from the AAS submodel templates published so far, a set of test cases each consisting of request and expected result, and multiple mockup servers simulating assets for evaluating asset synchronization using different protocols. The test framework is reusable and can be applied to any new AAS implementation that might emerge. It can also be iteratively applied to the already tested AAS implementations as new features are realized. Second, we apply this test framework to available open-source implementations and present the results in a comprehensive manner enabling easy comparison of different implementations so that potential users can determine which implementation is best suited for their needs. By identifying similarities and differences between implementations we create valuable feedback for the standardization process of the AAS.

The remainder of this paper is organized as follows. Section 2 depicts the process used to identify and select relevant open-source implementations and introduces the implementations evaluated in the paper. In Section 3 the evaluation approach and criteria are presented, followed by the results of the comparison in Section 4. The paper closes with an analysis of related work in Section 5 and conclusions in Section 6.

## 2. Open-Source Implementations

In this section, we identify relevant open-source implementations of the AAS Type 2, also known as reactive AAS, meeting the following requirements:free and open-source,implementation of the AAS specification Part 1 (metamodel) [15],implementation of the AAS specification Part 2 (API), more precisely the RESTful version of that API [16], andsufficient documentation to be able to execute tests.

The first part of this section describes how relevant open-source implementations have been identified and selected for comparison. In the second part, the selected implementations are introduced in more detail.

### 2.1. Methodology and Selection

To identify relevant open-source implementations of the reactive AAS we used a structured process as depicted in Figure 3. To cover both implementation- and academia-driven implementation we are using GitHub, the self-proclaimed “largest open-source community in the world” [17], and Google Scholar, the presumably largest database of scholarly information [18], as data sources.

With GitHub, we used the search term *Asset Administration Shell* which yielded 50 results. With Google Scholar, we used the search term *Asset Administration Shell AND implementation* and a time filter ≥ 01.01.2018 (as the relevant AAS specification has been published shortly before that). To limit the number of results we only considered the first 50 results ordered by relevance. We then scanned these 50 papers for URLs of source code repositories which returned 15 code repositories. Merging the results from GitHub and Google Scholar and removing duplicates resulted in 58 unique code repositories related to AAS. Of these 58 repositories, only nine implement the AAS Type 2 while the others either implement only the AAS Type 1 or contain other AAS-related software components or demonstrators. As a last step, we did filter out repositories that have not been active in the last 12 months because the AAS specification is currently rather new and still under constant evolution so any implementation that has not been updated for over a year does not hold much value for the future. This left us with the six repositories listed in Table 1.

After investigation, we decided to additionally eliminate two of these implementations for the following reasons:dfkibasys/asset-administration-shell: because it is an extension with very limited features based on the well-established Eclipse BaSyx implementation already present in the list, andJMayrbaeurl/opendigitaltwins-aas-azureservices: as the documentation is inadequate and this software therefore could not be started and tested by us.

This leaves us with a total of four implementations: AASX Server, Eclipse BaSyx, FA^3^ST Service, and NOVAAS.

### 2.2. Selected Implementations

In this section, the remaining four implementations are introduced and their characteristics are briefly explained.

#### 2.2.1. AASX Server

The AASX Server [19] is being developed in the context of the IDTA. Its code is based on the AASX Package Explorer [20] which is the most prominent modeling tool for AAS. AASX Server is implemented in C# .NET and comes in three variants:**core** containing only the server with a command-line interface (CLI) using .NET Core,**blazor** also containing a graphical user interface (GUI), and**windows** for running on windows without administrator privileges using .NET Framework.

#### 2.2.2. Eclipse BaSyx

Originating from the BaSys 4.0 and the follow-up BaSys4.2 research projects funded by the German Federal Ministry of Education and Research (BMBF), Eclipse BaSyx [21] provides an implementation of the reactive AAS (Type 2) [22,23]. Although Eclipse BaSyx is available for multiple programming languages (Java, C#, and C++) we will only investigate the Java version in this paper as it is the most mature and feature-complete of the three implementations. Eclipse BaSyx provides a feature-rich ecosystem including a client SDK as well as components for asset integration and AAS visualization. It is published under an MIT license and is well-known and often used in the AAS community.

#### 2.2.3. FA^3^ST Service

FA^3^ST Service [4,24] is being developed by the authors of the paper at Fraunhofer IOSB as part of the Fraunhofer Advanced Asset Administration Shell Tools for Digital Twins (FA^3^ST), a collection of tools for modeling, creating and using DTs based on the AAS specification. It is published as open source under Apache 2.0 license. FA^3^ST Service focuses on AASs on the edge, meaning synchronizing assets and AAS is a central aspect. It also focuses on easy usage for non-expert users while at the same time offering an open architecture that allows for easy extension of functionality.

#### 2.2.4. NOVAAS

The NOVA Asset Administration Shell (NOVAAS) [25,26] is being developed by NOVA School of Science and Technology in the context of the H2020 PROPHESY project [27]. The implementation follows a no/low-code approach based on Node-RED, a flow-based programming tool, and is published under EUPL v1.2 license. Besides the AAS metamodel and API implementation, NOVAAS offers a web-based graphical user interface that allows the easy creation of dashboards for non-expert users.

## 3. Evaluation Approach

To evaluate and compare the different AAS Type 2 implementations regarding their supported features and usefulness for creating AAS-based applications we define a set of criteria. These criteria are introduced and explained in detail in Section 3.1.

Although most of these criteria have been evaluated based on documentation and the code itself, we also conducted tests to verify the supported AAS metamodel elements and API functionality. A detailed description of the test framework is presented in Section 3.2.

### 3.1. Evaluation Criteria

In this section, the evaluation criteria for comparing different implementations are introduced. These criteria have been selected based on multiple aspects: relevant issues when using open-source software, aspects related to compliance with the AAS specification, and technical requirements when working with DTs that go beyond the AAS specification. The criteria can be divided into the following groups: general criteria, AAS-specific criteria, implementation-specific criteria, and ecosystem criteria.

#### 3.1.1. General Criteria

The general criteria focus on aspects related to open-source software and typical usage scenarios.**Main Contributors**: Which organizations are the main contributors driving the implementation? We consider an organization a main contributor if it provided at least five commits and ten thousand lines of code.**License**: Under which license is the implementation published? The license affects which rights and obligations a user has and whether it is compatible with its own development guidelines.**Programming Language**: Which programming language(s) does the implementation use?**Usage**: What types of usage does the implementation support? Three typical types of usage are considered: via a command-line interface (CLI), as a docker container, or as an embedded library from custom code.

#### 3.1.2. AAS-Specific Criteria

This group of criteria focuses on to which degree the implementations realize the functionality defined in the AAS specification.**Metamodel Version**: Which version of the AAS metamodel (as defined in the AAS specification Part 1) is supported by the implementation? Since 2018, at least three versions of the specification have been published [2].**Model Formats**: According to AAS specification Part 1, the AAS metamodel can be serialized using multiple different serialization formats. This criterion describes which of these serialization formats is supported by an implementation. Support for different serialization formats is relevant for two aspects: importing an existing AAS model upon start-up as well as for communication via HTTP although the AAS specification defines JSON as the only supported serialization format for communication via HTTP. This criterion therefore primarily focuses on the supported serialization formats for the import of existing AAS models at start-up. As the serialization formats AutomationML (AML) and OPC UA NodeSet are not developed by the authors of the AAS specification but rather standardization bodies related to AML resp. OPC UA, standardization of these serialization formats is not published as part of the AAS specification itself. Therefore, we only consider AASX, JSON, XML, and RDF in this paper.**API Interfaces**: The AAS specification Part 2 defines as set of so-called *Interfaces* consisting of *Operations* describing the AAS API in a protocol-agnostic manner. This criterion describes to which degree these interfaces and operations are supported by an implementation. The interfaces *AAS Registry* and *Submodel Registry* are excluded from this list as they are typically implemented as separate applications. They are therefore later introduced as the ecosystem criteria *Registry*. For some of these interfaces, we evaluate these criteria based on actual tests described later in this section. The rest is evaluated based on documentation.**API Protocols**: Which protocols can be used to communicate with the AAS via its API? So far, there are proposals on how to map the protocol-agnostic interfaces and operations to HTTP and OPC UA, although no official final version of these mappings has been released.

#### 3.1.3. Implementation-Specific Criteria

This group of criteria focuses on aspects that are related to implementation details including aspects tightly related to the AAS specification but still beyond what is defined in the specification itself such as synchronization of assets with an AAS, security, or how data can be persisted.**Asset Synchronization**: Synchronizing the (physical) asset(s) with their digital representation, the AAS, is an essential requirement when working with DTs. However, this aspect is not covered by the AAS specification at all. We introduce this criterion, as from our perspective this is an essential capability of a DT. This criterion includes asset connectivity regarding three well-established communication protocols, HTTP, MQTT, and OPC UA, combined with the four interaction patterns for DTs, read, write, execute, and subscribe.

In theory, all implementations can be made to support all protocols and interaction patterns via manual coding. However, this criterion does not try to evaluate if it is possible at all to synchronize with an asset but rather if there is any user-friendly or predefined way to do so. In the best case, this can be done via configuration without writing any code. We also consider this criterion (partially) fulfilled if there is a mechanism available to integrate custom code, e.g., a software interface, or at least an example of how to do this is provided.

**Data Persistence**: An AAS Type 2 implementation allows modifying the hosted AAS model at run-time which raises the question of how these data can persist by the implementation. This criterion describes three possible forms of persistence: in-memory (meaning all changes are lost if the software is stopped), file-based (meaning the AAS model is serialized to a file according to one of the supported serialization formats), and database-based (meaning the data persist in a database). Often implementations support multiple of these types of persistence and allow switching between them with little effort.**Security**: Similar to asset synchronization, security is not (properly) addressed in the AAS specification. Nevertheless, as it is an essential aspect when using any implementation in real-world applications, we introduce this as a separate criterion. Possible implementations may greatly differ in complexity and the level of security they provide, especially when it comes to access control. In this paper, the primary aspect is if the implementations offer any solution to security at all and not the level of maturity.**Other Features**: This criterion shall be used to present additional features of an implementation that seem relevant and do not fall into the above categories.

#### 3.1.4. Ecosystem Criteria

DTs follow a complex life cycle involving multiple phases: identify, model, develop, share, use, and validate [28], of which an AAS Type 2 implementation only realizes the *develop* phase. Therefore, it is not uncommon that these implementations are part of a bigger framework or set of tools that provide support for other phases of the DT life cycle such as modeling the DT. With this group of criteria, we evaluate what type of related software artifacts relevant for working with DTs are published either as part of the implementation or alongside it.**Client Library**: Connecting applications to deployed Type 2 AAS is essential. This requires applications to know and “speak” the AAS APIs. A client library offers an easy-to-use software library that hides the complexity of the AAS API and therefore simplifies and accelerates the development of applications using the DT. However, it typically needs multiple client libraries targeting different programming languages.**Model Editor**: Modeling a DT can be a complex task and is often not done by software developers familiar with the different serialization formats of the AAS. A model editor simplifies that task by offering a graphical user interface to easily create and modify the AAS model also for non-experts.**Model Validation**: Does the implementation or framework verify the compliance of a given AAS model with the specification including all constraints defined? This primarily focuses on a separate explicit validation step or tool that is executed automatically or manually and provides clear indicators of how the model violates the constraints. It is noticed that every implementation inherently has some sort of implicit validation in the form of crashing or throwing an exception when called with an invalid model. However, these implicit validations do typically not provide detailed and helpful information on the cause and leave a user puzzled about how to fix the model. Therefore, this criterion is only fulfilled if an implementation provides at least some form of explicit validation.**Registry**: Although the interfaces *AAS Registry* and *Submodel Registry* are defined in the AAS specification Part 2 together with all the other interfaces, they are not considered part of an AAS Type 2 but rather a tool on its own.**Visualization**: By visualization, we refer to some sort of graphical user interface visually depicting the current state of the AAS. Visualizations can come in many forms, e.g., depicting the AAS model structure or displaying a single value over time as a graph. Visualizations are sometimes combined with a model editor allowing the modification of the AAS model at run-time.**Other**: What other relevant, interesting, and/or unique tools are part of the ecosystem?

### 3.2. Test Framework

This section introduces the test framework used to perform actual tests against the different implementations. The tests include a limited subset of the AAS API functionality as well as asset synchronization for some selected protocols. In the remainder of this section, the AAS model used as the basis for the tests is introduced followed by a detailed description of the use cases.

#### 3.2.1. AAS Model

Testing the API requires a suitable AAS model. Since the AAS specification defines many types of submodel elements, we tried to limit our test model to a reasonable subset of these submodel element types. For this, we analyzed all published submodel templates [29] for the submodel element types used and decided to test each submodel element type used in at least one of the submodel templates. The result of this analysis is shown in Table 2.

A simplified version of the AAS model used for testing is shown in Figure 4 depicting the overall structure including types and idShort of the elements. The model contains a single AAS with four submodels. The submodel *ElementTypes* contains all the different types of submodel elements identified in the previous step. It also includes a submodel element collection containing a property as this is relevant for testing the output parameter *level*. The other three submodels are related to testing asset synchronization; one submodel for each network protocol tested. Each submodel contains submodel elements according to the capabilities of the related protocol, e.g., a property for read/write and/or for subscribe, and an operation to be executed on the asset if this is supported by the protocol.

That AAS model was created using AASX Package Explorer and exported as AASX, JSON, and XML. As the AASX Package Explorer only supports v2 of the AAS metamodel we used the FA^3^ST Package Explorer Converter [30] to convert the model to v3.

#### 3.2.2. Test Cases

In theory, testing multiple different implementations of the same specification should be straight forward. Based on this assumption, our initial plan was to conduct a test of the complete API for each implementation. However, we soon discovered that, unfortunately, no two implementations implement the API in the same way. To our understanding, this is mainly for the following reasons. First, there is no stable/final HTTP mapping of the AAS API published yet. On the contrary, there are multiple different release candidates and intermediate draft versions publicly available. This does not only apply to the HTTP mapping of the AAS API but also to the AAS metamodel, for which the latest stable version published is v2 but several release candidates have been published for v3.

Some of the implementations considered in this paper seem to be currently developing newer versions of their software based on metamodel v3 and newer HTTP API definitions but are only going to release them once the underlying specifications are finalized and published. However, we are only considering released or, if not available, at least relatively stable versions of the implementation with minimal documentation, which does not include these development branches.

##### API Interfaces

We based our tests on the AAS interfaces and operations defined in the AAS specification Part 2 v1.0RC02 [16] and the corresponding Swagger documentation labeled “Entire Interface Collection v1.0RC01” [31]. As each implementation uses a (slightly) different API which prevents using a common test to evaluate all of them and instead needs to manually convert each test case to each of the implementations’ APIs, we decided to limit the tests to a somewhat representative set of API calls. The test, therefore, includes operations for the three interfaces AAS Repository, AAS, and Submodel as these are the most important interface without an AAS implementation that is hardly usable in real-world applications. For the AAS Repository interface creating, reading, updating, and deleting AAS is tested. For the AAS interface, reading the AAS and its contained submodel references is tested as well as adding a new submodel For the Submodel interface, reading and updating submodels and submodel elements is tested as well as creating, updating, and deleting elements. Output modifiers are tested only for a single API call assuming that if an implementation provides the functionality to process an output modifier for one API call, they also do it for all others as the hard part with output modifiers is the implementation of the underlying logic.

##### Asset Synchronization

Although asset synchronization is not part of the AAS specification we think this an essential capability of a DT and therefore decided to test how well this is supported by the implementations.

Asset synchronization can comprise up to four different types of interactions depending on the communication protocol used by the asset: read, write, execute (an operation), and subscribe. Figure 5 shows a protocol- and implementation-agnostic sequence diagram of how the test framework tests these different types of interactions. For testing the execution of an operation, we assume an operation that takes two numbers as input parameters and returns the sum of them as a result. The italic calls between AAS implementation and asset are protocol-specific and hence here only depicted generically.

The test framework includes tests for three communication protocols: HTTP, MQTT, and OPC UA. These were selected because they are some of the most widely used network protocols in industry and are the only network protocols mentioned in the AAS specifications. For each of the three protocols, we created dummy or test assets providing just enough functionality to test the different interactions. Table 3 gives a detailed view of the protocol-specific APIs of these test assets. The cells in this table are the protocol-specific realizations of the protocol-agnostic messages depicted in italics in the sequence diagram. Not all protocols support all interactions, e.g., HTTP does not support subscribe and MQTT does not support read and execute.

## 4. Evaluation Results

We evaluated the four open-source AAS Type 2 implementations identified in Section 2 based on the criteria introduced in Section 3. An overview of the results is shown in Table 4. Most criteria are formulated as binary questions, meaning whether the implementation fulfills the criteria or not, where yes is represented by a check (✓) and no by a dash (-). In some cases, a more differentiated answer in the form of ‘yes, but…’ is required. These cases are represented by a check in brackets and are discussed in detail in Section 4.1.

Although most of the criteria have been evaluated based on documentation or high-level analysis of the actual code, some aspects of the API have been tested. The overview of the test results is shown in Table 5.

### 4.1. Detailed Results

In the following, the results of Table 4 and Table 5 are explained in more detail per implementation.

#### 4.1.1. AASX Server

AASX Server implements a rather limited subset of the AAS API and therefore seems a bit outdated. The latest release (*AASX Server 2022-07-25.alpha*) is still an alpha version and, at the time of writing, is over six months old. The repository contains lots of branches that seem to indicate that development is still ongoing including major changes such as adaptation of the v3 AAS metamodel and newer versions of the AAS API. However, in this paper, we only evaluated the latest released version or, if there has not been a release so far, the code available on the main branch of the repository. The implementation offers an easy-to-use command-line interface and docker container and configuration via command-line arguments.

AASX Server does not claim to provide an AAS-compliant REST API as indicated by a comment in the source code: “Please notice: the API and REST routes implemented in this version of the source code are not specified and standardized by the specification Details of the Administration Shell. The hereby stated approach is solely the opinion of its author(s)” [32]. Nevertheless, many operations of the AAS API specification are supported, but often with a slightly different syntax. For example, each call uses a different URL pattern, sometimes the return code is different, sometimes a different HTTP method is used, and the JSON payload usually has a slightly different format. Additionally, instead of base64URL-encoded identifier (globally unique), idShort (not globally unique, only within the parent element) is used without any encoding for addressing AASs or submodels.

AASX Server uses v2 of the AAS metamodel resp. v3 in the undocumented API. Regarding supported model formats, AASX Server supports JSON only as a payload format for API calls and XML only in combination with AASX, meaning that only AASX files using XML can be loaded at start-up.

AASX Server comes in three variants: *blazor*, *core*, and *windows* (see Section 2.2.1). According to the documentation, all variants should offer the same API, e.g., the description of the *blazor* variant reads “This variant uses Blazor framework to provide a graphical user interface in the browser for exploring the AASX packages. The other APIs are the same as in the core variant” [19]. After receiving feedback from the authors, it came to our knowledge that the *blazor* variant does offer an additional API the implements more features than the default one. Since this API is only available in one of the three variants and users are probably not going to find it because of missing documentation, we decided to keep the evaluation results of the documented API. Nevertheless, we repeated our tests for the newer, undocumented API and included these results wherever they deviate from the documented API.

Regarding API interfaces, the documented API of the AASX Server supports the operations related to reading data of the AAS interface but not the ones related to creating, updating, or deleting elements. The Submodel interface is partially supported as AASX Server does not support reading a list of all submodels elements of a submodel, creating (POST) and updating (PUT) are combined in one call, invoking operations is not supported, and custom names for output modifier (extent, level, content) are used without documentation leaving the user unclear about the provided functionality. For the documented API, the AASX File Server interface supports read and update but not delete. The undocumented API fully supports all interfaces besides Descriptor.

Although AASX Server supports both HTTP and OPC UA as API protocols and can provide both at the same time, they are not synchronized, i.e., changes made via one of the APIs are not reflected via the other, which renders this feature useless in real-world applications.

AASX Server provides limited support for asset synchronization. It supports reading values from an OPC UA-based asset as well as subscribing to value changes. However, subscribing is limited by the fact that the node IDs on the OPC UA server representing the asset must follow a given naming scheme that restricts usage in brownfield scenarios which are typical in real-world applications. Furthermore, subscriptions are implemented as a periodically executed read operation. Configuration of the asset connection happens via the AAS model itself using *qualifiers*. The documentation provides minimal information on how to connect to an asset, so it was necessary to inspect the code to obtain the relevant information.

According to the documentation, AASX Server does implement some sort of security, but no details are provided on what this entails or how it works. Analyzing the source code indicates that some form of authentication and authorization happens for accessing the API via both HTTP and OPC UA.

From an ecosystem perspective, AASX Server seems to do rather well at first glance including a C# client library, the model editor, a registry implementation, and a visualization. However, AASX Client [33] and AASX Registry [34] have both not been updated in over two years, have never been released and have minimal to no documentation. A visualization of the AAS is available via the *blazor* version. The model editor aspect is addressed by the AASX Package Explorer [20] which integrated the AASX Server and allows the running of the AASX Server directly from within the editor view. The AASX ecosystem additionally includes a Python-based tool to convert XLS to AASX files [35].

The tests have been conducted with the *core* variant (for the documented API) and the *blazor* variant (for the undocumented API) of the *AASX Server 2022-07-25.alpha* release [36].

#### 4.1.2. Eclipse BaSyx

Eclipse BaSyx is a well-established open-source project providing different tools and components along the DT life cycle for different programming languages. In this paper, we only evaluate the Java version. Similar to the AASX Server, Eclipse BaSyx is working on an updated version of their software which might include additional features, but has not been evaluated in this paper as it has not yet been released yet [37,38].

Eclipse BaSyx is divided into several components and services such as AAS Server, AAS Registry, or AAS Web Client, that can be started in different combinations as required via CLI, docker, or embedded libraries. Configuration is done via multiple configuration files or environment variables.

Eclipse BaSyx supports loading models in the data formats AASX, JSON, and XML. It also supports almost all operations of the three most important API interfaces: AAS Repository, AAS, and SM. The operation *PostAssetAdministrationShell* is marked as (✓) in Table 5 because the functionality is implemented but not as defined by the API, i.e., using PUT instead of POST and returning status code 200 with no content instead of 201 containing the newly created AAS. Eclipse BaSyx does not use base64URL-encoding identifier in the HTTP paths. Although the operations *GetAssetInformation* and *PutAssetInformation* of the AAS interface are not supported we decided to state that Eclipse BaSyx fully implements this interface as this is a consequence of using v2 of the metamodel which does not specify asset information (it was only introduced in v3). Furthermore, it supports only the value-only content format which affects both the AAS interface as well as the Submodel Interface. OPC UA is not supported as an API protocol.

Eclipse BaSyx offers multiple ways to synchronize with an asset: via custom coding, via a component called DataBridge, or via a mechanism called property/operation delegation. Via custom coding, theoretically, any kind of asset can be connected to the AAS; however, this requires manual writing of the code to do so. The DataBridge is a component external to the AAS Service that helps to synchronize with assets by subscribing to changes on the asset and forwarding them to the AAS via HTTP commands. It is based on Apache Camel, a message-oriented middleware for integrating systems producing or consuming data, which already supports many different types of data sources such as HTTP or MQTT. The delegation mechanism allows reading property values and executing operations via HTTP and is configured by adding qualifiers to the corresponding element of the AAS model.

Eclipse BaSyx supports storing its data in memory, in a MongoDB database, or on the file system (which requires minimal coding) and offers a rule-based security approach that allows assigning access rights on a per resource level. It also supports publishing changes to the AAS and repository structure as well as value changes via MQTT. Via so-called feature decorators, Eclipse BaSyx provides a way for easy extension of functionality.

The BaSyx ecosystem includes a client library, registry, and visualization in the form of the AAS WebGui, as well as a limited model validation that does basic sanity checks but does not cover all constraints defined in the AAS specification. Additional rather unique components are the DataBridge and a UI plug-in mechanism that allows developing custom plug-ins to display submodels and submodel elements in certain ways depending on their semantic id. Eclipse BaSyx also provides an easy way to integrate KeyCloak as an identity and access management solution with minimal custom coding required.

The tests have been conducted using v1.3.0 of the Eclipse BaSyx Java SDK [39] and Eclipse BaSyx Java Components [40]. Because of the issue with unsupported URL-based identifiers the test model had to be manually updated.

#### 4.1.3. FA^3^ST Service

FA^3^ST Service is an AAS Type 2 implementation with a focus on ease of use and asset synchronization. The architecture defines multiple software interfaces, e.g., for persistence or asset connection, which enables easy extension and customization. Configuration is done via a single configuration file allowing the choice and configuration of which implementation(s) of these interfaces to use. FA^3^ST Service also pays special attention to asset synchronization by including the concept of asset connection directly in the implementation and providing out-of-the-box support for different protocols that can be used without writing any code. It can be used via CLI, docker, or as an embedded Java library and offers multiple means for configuration (file, command-line arguments, environment variables).

FA^3^ST Service is the only implementation of the four considered in this paper that implements v3 of the AAS metamodel. It supports loading data from all four data formats and supports most of the API interfaces, including most of the output modifiers (except content = metadata). Furthermore, it supports both API protocols, HTTP and OPC UA.

FA^3^ST Service supports all criteria related to asset synchronization evaluated in this paper. Regarding data persistence, FA^3^ST Service only supports storing data in memory or in a file. It also does not provide any type of security other than being able to communicate with assets securely. A unique feature of FA^3^ST Service is the capability of hosting the API both via HTTP and OPC UA at the same time while synchronizing them, meaning that changes via one are immediately reflected via the other providing a consistent state across different protocols. FA^3^ST Service also stands out with its open architecture allowing for easy extension or customization of many aspects of the software without the need to recompile FA^3^ST Service itself, e.g., for adding support for additional communication protocols for the asset connection.

Although defining the vision of a complex ecosystem [28], FA^3^ST is currently lacking major tools such as client libraries, registry, or visualization. However, FA^3^ST Registry as well as visualization are currently being developed. At the time of writing, the FA^3^ST ecosystem offers the FA^3^ST Package Explorer Converter [30], a tool that converts AAS models designed with the AASX Package Explorer to be used with FA^3^ST and the EDC Extension for AAS [41] that allows integrating AASs into the Eclipse Dataspace.

The tests have been conducted using v0.4.0 of the FA^3^ST Service [24] via CLI.

#### 4.1.4. NOVAAS

NOVAAS is implemented using the flow-based visual programming tool Node-RED which itself is based on JavaScript. Its primary intended and only documented form of usage is via docker.

NOVAAS uses v2 of the metamodel and has limited model format support. It supports loading data from AASX files that are based on JSON and JSON as API payload format. The supported API interfaces are AAS Repository, AAS, Submodel Repository, and Submodel, whereby implementation is partially incomplete. For the AAS Repository interface, NOVAAS only supports the read operations. For the AAS interface updating an AAS is not present as well as operations related to asset information (although this is because NOVAAS uses v2 of the metamodel that does not define asset information which is the same for AASX Server and Eclipse BaSyx). Regarding the Submodel interface, all operations but the operation GetAllSubmodelElements are implemented. However, fetching the submodel itself typically includes this information. NOVAAS is the only implementation compared in this paper that supports the output modifier content = metadata. Similar to Eclipse BaSyx, NOVAAS does not use base64URL-encoded identifiers.

Asset synchronization in NOVAAS is a bit different from other implementations because it is based on Node-RED. NOVAAS offers a predefined extension point with the Node-RED flow to synchronize with assets via the *read* interaction pattern and provides examples of how to use it with HTTP [42] and OPC UA [43] protocol. Although NOVAAS uses a database for storing historical values, it does not store the AAS model in the database and therefore only supports in-memory persistence according to the definition of persistence used in this paper. In the tested version, the only security feature offered by NOVAAS is limiting access to the user interface (UI) by a simple username/password prompt. However, there exists a version of NOVAAS that provides additional role-based security for accessing the UI and API [44]. Besides the unique feature of using a no/low-code approach, NOVAAS can publish value changes via MQTT.

The NOVAAS ecosystem is rather limited and does not include a client library, model editor, model validation, or registry. It does include a web-based visualization featuring customizable dashboards and statistics as well as an integration with KeyCloak for managing access rights to the web front.

The tests have been executed using the published docker container with tag *87e613ef727eab44502caa552b08ff019c8fc908* (https://registry.gitlab.com/novaas/catalog/nova-school-of-science-and-technology/novaas:87e613ef727eab44502caa552b08ff019c8fc908, accessed on 12 April 2023).

### 4.2. Discussion

The results show that there is no AAS Type 2 implementation that implements the AAS specification in all details. Most likely this is because there is no official stable version of the HTTP API released yet but only slightly different proposals or release candidates. The degree to which the implementations support the specification and particularly the HTTP API varies substantially. It seems logical that this is related to the HTTP API specification not yet being stable. This assumption is also substantiated by the fact that most implementations are still under development and often already have some feature branch(es) implementing more functionality or newer versions of the HTTP API. All implementations can be used as docker containers while most also provide a CLI and only half of them can be used as embedded libraries.

Regarding AAS-specific criteria, most implementations currently support v2 of the metamodel while most are working on support for v3. Although AASX is defined as the primary model format for exchanging AAS models it is not fully supported by one of the implementations (NOVAAS, which only supports AASX in combination with JSON but not XML). Only two out of the four implementation support loading AAS models from JSON files (Eclipse BaSyx and FA^3^ST Service). This seems to be a missed opportunity as all implementations do have the basic ability to deserialize JSON as it is used as a payload format for the HTTP API. The same applies to XML support in the AASX Server as supporting AASX requires XML deserialization to be implemented. RDF model format is only supported by FA^3^ST Service.

The implementations vary substantially in number and completeness of and compliance with the API interfaces realized. The most important interfaces for most uses, AAS repository, AAS, and Submodel, are supported by all implementations at least to some degree. Other interfaces are implemented by only two implementations (CD repository, AAS Basic Discovery, and AASX File Server) or even none of them (Descriptor).

Regarding the AAS protocols, all AAS implementations support HTTP, while only two support OPC UA (AASX Server and FA^3^ST Service). However, only one AAS implementation (FA^3^ST Service) supports synchronizations between the two AAS protocols, although this is an expected functionality. The supported protocols affect the applications that AASs are intended to realize, and the expertise required to develop and use them. For example, one might expect that there would not be as many applications with AAS at the shop-floor level, where OPC UA is a typical protocol and not HTTP.

Regarding asset synchronization, implementations seem to either provide quite sophisticated support (Eclipse BaSyx and FA^3^ST Service) or only very limited (AASX Server and NOVAAS). Although Eclipse BaSyx and FA^3^ST Service both support all modes and protocols tested they do have different approaches to asset synchronization. Eclipse BaSyx itself offers two approaches to asset synchronization: via writing custom code (which works for any type of interaction and any network protocol) or via DataBridge, a separate software component that can subscribe to changes in the asset(s) and update the AAS accordingly. FA^3^ST Service also has a two-fold approach. It provides a simple software interface that allows implementing asset connectivity via custom code, but it also provides predefined implementations for multiple protocols that can be used via configuration files without the need to write any code.

Even though the asset synchronization functionality is not addressed by the AAS specification, it is mandatory for DTs. Realizing this functionality will ensure that a physical asset and its digital representation are synchronized bidirectionally with a certain degree of synchronization (frequency and accuracy). Once this functionality is fully implemented for typical communication protocols and interaction paradigms, the applicability of AAS will be greatly enhanced, and more complex use cases can be supported that go beyond basic data acquisition and lead to closed-loop applications.

Regarding data persistence, all implementations support in-memory persistence, i.e., keeping the data only in the main memory. For AASX Server and NOVAAS, this is the only type of persistence supported. However, the drawback of this type of persistence is that all changes are lost once the application is stopped. Although not able to retain AAS model information, NOVAAS uses a database to store the historical values of properties. Both Eclipse BaSyx and FA^3^ST Service support one type of persistence, either file- or database-backed, that keeps the application state when stopped the service is stopped.

Security support varies substantially across the implementations. From not offering any form of security (FA^3^ST Service), through (role-based) authentication and authorization for accessing the API (AASX Server and NOVAAS) to access rights on a per resource level (Eclipse BaSyx). This heterogeneity related to security is quite expected, as security is not yet fully addressed in the AAS specification. This will change in the future, as plans have already been made to work on this and a working group has been established. A draft version has been part of an earlier version of the AAS specification but has been withdrawn until the specification is sound.

The most common additional feature is the ability to publish events about changes to the AAS model via MQTT. This is a feature discussed to become part of the AAS specification but not yet standardized.

The ecosystems around the different implementations are respectable although essential tools such as client libraries, model editor, or model validation are rarely available. However, many tools are currently under development related to the different implementations and will likely become available soon. In an ideal world, different tools implementing the same specification would be interoperable. Unfortunately, this is currently not the case as none of the implementations is fully compliant with the specification. As the stable v3 of the specification is planned to be released in the first half of 2023 this hopefully will change soon and the tools of the different ecosystem will become interoperable.

Based on the above discussion, one could conclude that the more completely the AAS specification is covered by AAS implementations, the more use cases are supported, and the greater the impact. Furthermore, only if AAS implementations implement the same version of the AAS specification, and without “interpreting” some of its aspects, the I4.0 vision of interoperability could be fully realized and vendor (or even implementation) lock-in avoided. In the future, it is expected that a typical use case will contain many AASs, each of which can be implemented by any AAS implementation, but will be interoperable by default without any development or even configuration.

However, this requires the more coordinated efforts of both the teams implementing the AAS specification and the AAS standardization body coordinating the AAS specification. Although the AAS teams discover problems with the AAS specification during its implementation, the AAS standardization body should take their feedback into account to improve the usability of the AAS specification. This has already been happening within the IDTA, and it is expected that cooperation will continue to be strengthened in both directions in the future.

## 5. Related Work

AASs have been a very important topic in recent years and many papers have applied AASs to realize Industry 4.0 use cases while maintaining interoperability. However, there are not so many papers that compare the existing AAS implementations.

In [45], the authors discussed the current state of AASs, and the frameworks used for them. The focus is on two AAS frameworks, namely Eclipse BaSyx and the AASX Server, which according to the authors are predominantly used for the AAS use cases. This conclusion was drawn based on the framework used to develop the AAS demonstrators for Hannover Messe 2022.

The paper analyzes how these two frameworks deal with the communication between AAS and the asset. The authors found several problems with these implementations. With respect to AASX, for example, they found that MQTT communication is not event-based, but is triggered by a fixed timer, and the endpoints are not synchronized. To address these issues, the authors propose an event-based solution to reduce response times between assets and AASs.

In this paper, rather than taking a subjective approach, we have taken an in-depth approach to the selection of AAS implementations to be compared. In addition, the selected AAS implementations were compared not only in terms of how they implement asset connectivity, but also based on many other criteria that combine both the details of the AAS specifications and the well-known practices of open-source development.

In [46], the authors intentionally developed AASs using different tools. The tools considered were the AASX Server and Eclipse BaSyx. The selection of the tools was based on the authors’ subjective estimation of the tools’ level of recognition in the community and the projects’ maturity levels. All the implementations were done by the same developer. The authors found the AASX Server to be easier to use than the Eclipse BaSyx. In the future, they plan to test FA^3^ST Service as well.

Although the authors of [46] selected the tools according to their subjective assessment, the selection of tools in this paper is based on objective criteria. Furthermore, the tools themselves were not compared. Rather, the authors only explained the steps they took to implement the case study. In contrast, we compared the selected tools based on many criteria without considering a specific use case.

In [4], the authors introduced the FA^3^ST Service and in particular the FA^3^ST approach for asset connection. They presented other open-source projects related to the AAS specification such as AASX Package Explorer and AASX Server, Eclipse BaSyx, and Eclipse AAS Model for Java. However, they confirmed that the list of implementations they considered was not complete and that the selection was subjective. Additionally, the authors made a brief comparison with FA^3^ST Service, explaining only the main differences.

In contrast to [4], the selection of tools in this paper was based on objective criteria. Furthermore, we performed a detailed analysis of the selected tools and justified the results by creating a test environment with test cases applied to all implementations. Although the test environment allows the evaluation to be repeated, for example, for an additional tool, or for a new version of a tool already considered, the test cases and the results of their applications ensure the objective evaluation of the tools.

## 6. Conclusions

In this paper, we analyzed and compared the existing free and open-source implementations of the Asset Administration Shell (AAS) specification. To identify and select the AAS implementations to be compared, we performed a structured search using GitHub and Google Scholar as data sources. The results found were merged, duplicates were eliminated, and inactive implementations were filtered out. This process resulted in four implementations that were analyzed in detail.

To compare the different implementations, we defined a set of objective evaluation criteria. We also defined a test framework to test the support of the most common AAS model elements based on usage in published submodel templates and API calls. The test framework also includes the aspect of asset synchronization which is not part of the AAS specification but an essential functionality of any DT.

The evaluation approach was applied to the four selected AAS implementations. In Section 4, the results were summarized in a summary table as well as presented and discussed in detail for each implementation.

These results show that there is no AAS implementation that fully implements the AAS specification. There are some aspects of the AAS specification that are not covered by any implementation, and many that are not fully implemented. However, all considered AAS implementations can support the minimum required functions, such as reading and writing an AAS and communicating with it via the HTTP protocol. On the other hand, when it comes to synchronization with physical assets, many aspects are still open and should be addressed in the future.

This paper is the first attempt at a comprehensive comparison of AAS implementations. It gives hints on how these implementations could be extended and improved to be better than the others based on the missing or not fully implemented functionality. More importantly, it could be used to further refine the AAS specification to help software developers understand the semantics of the AAS metamodel and API.

This paper highlights two major aspects. First, the implementation of the AAS specification is still an open challenge, and second, there is a lot of incompatibility between different implementations. Presumably, the main reason for this is that no stable version of the AAS API has been released so far. However, this is expected to change soon as the release of v3 of the specification is planned for the first half of 2023.

This type of evaluation could be repeated once the new version of the specification is released, and the implementations had enough time to catch up with the implementation.

## Figures and Tables

**Figure 1 sensors-23-05229-f001:**
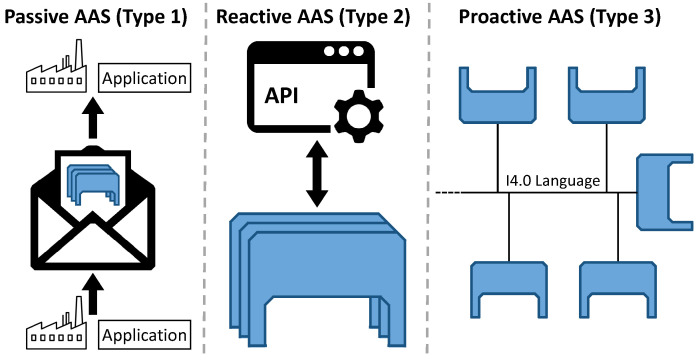
Different types of AAS. Reprinted with permission from [4]; 2022, IEEE.

**Figure 2 sensors-23-05229-f002:**
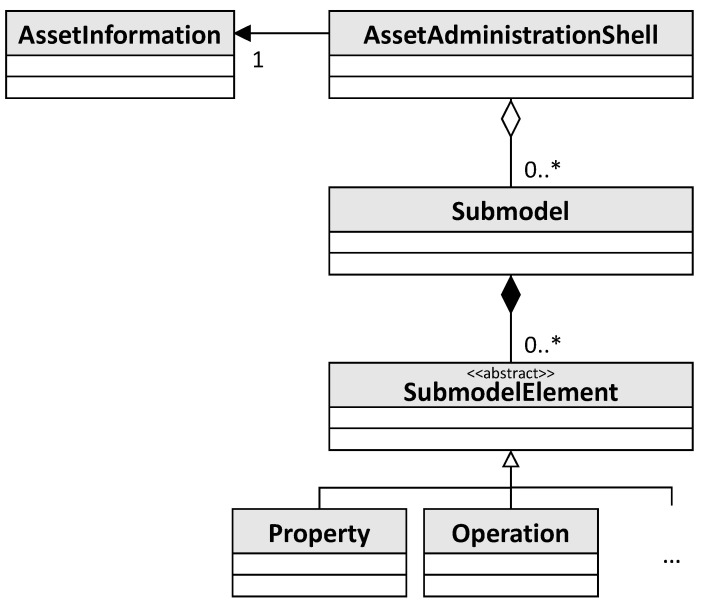
UML class diagram showing a simplified version of the AAS metamodel.

**Figure 3 sensors-23-05229-f003:**
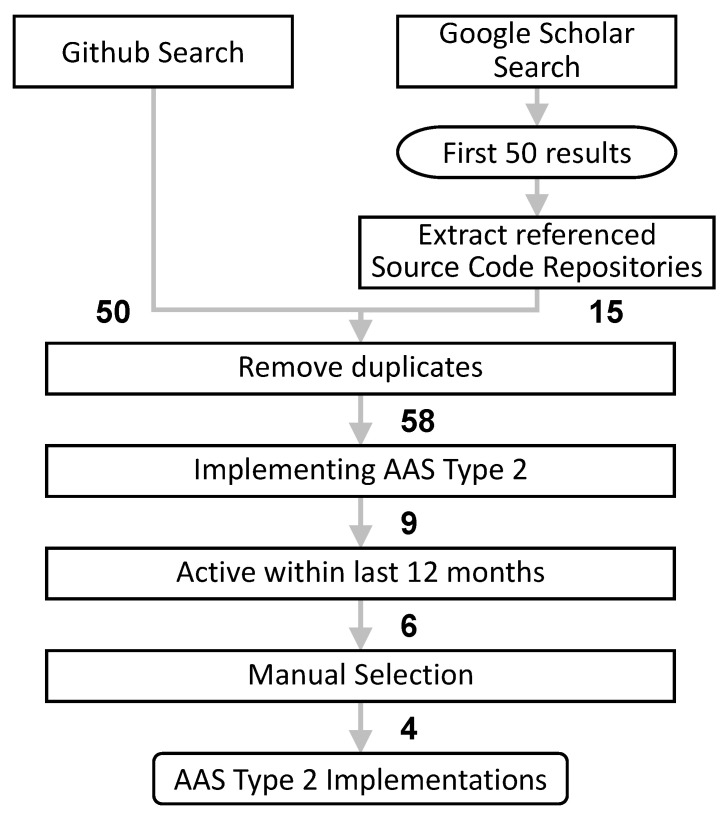
Selection process to identify relevant AAS Type 2 implementations. The numbers represent the remaining results at each step.

**Figure 4 sensors-23-05229-f004:**
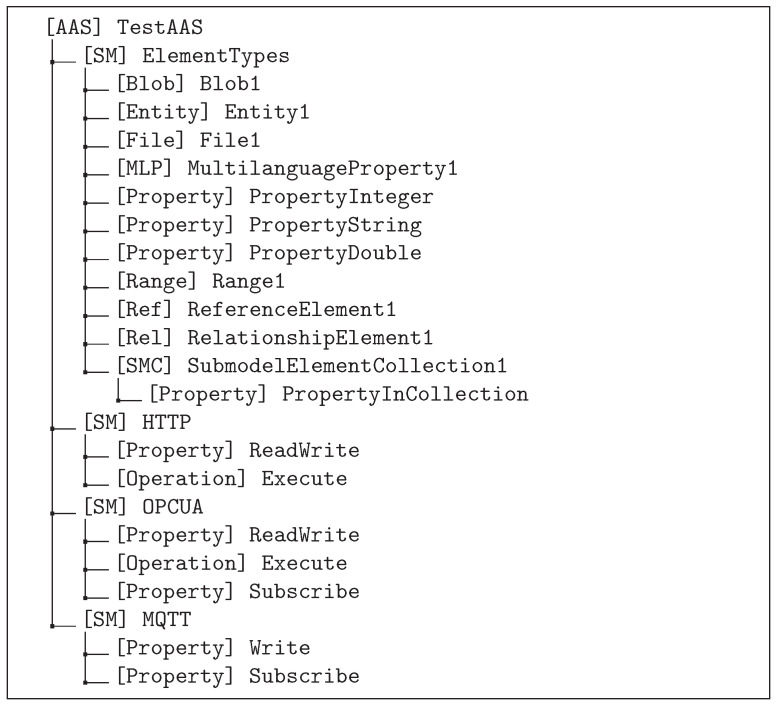
Simplified AAS model used for testing.

**Figure 5 sensors-23-05229-f005:**
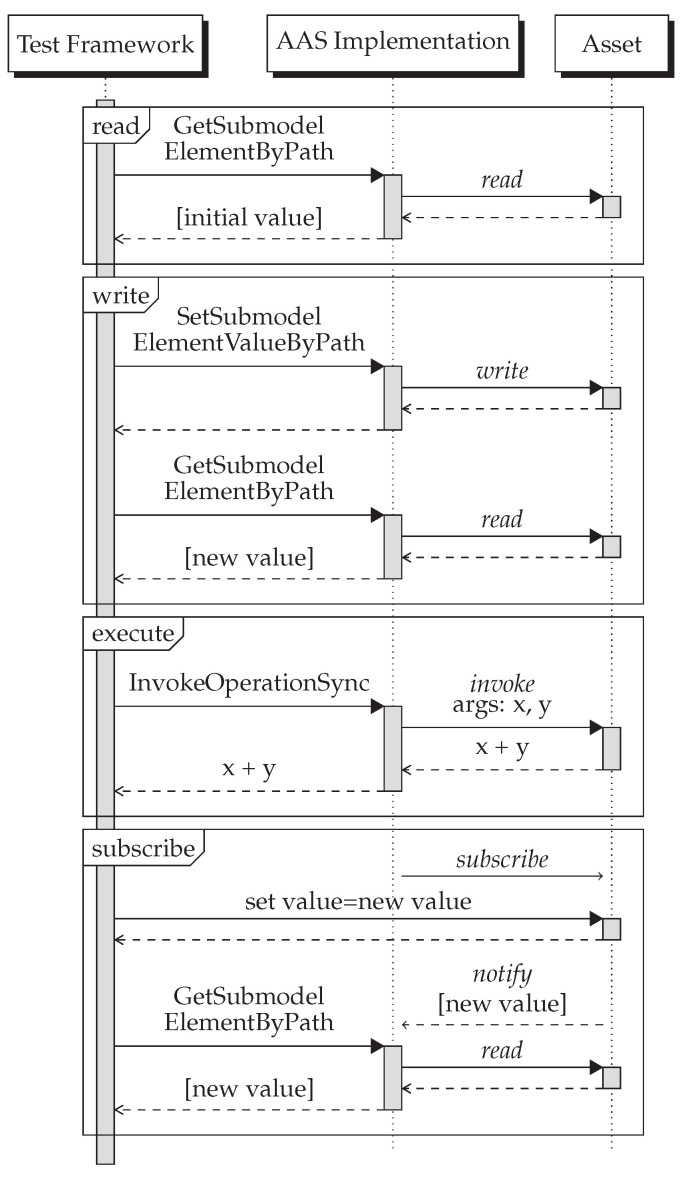
Sequence diagrams showing the protocol-agnostic test patterns for asset synchronization.

**Table 1 sensors-23-05229-t001:** Remaining repositories implementing AAS Type 2 (links last accessed on 12 April 2023).

https://github.com/admin-shell-io/aasx-server
https://github.com/dfkibasys/asset-administration-shell
https://github.com/eclipse-basyx/basyx-java-sdk
https://github.com/FraunhoferIOSB/FAAAST-Service
https://github.com/JMayrbaeurl/opendigitaltwins-aas-azureservices
https://gitlab.com/novaas/catalog/nova-school-of-science-and-technology/novaas

**Table 2 sensors-23-05229-t002:** Usage of different AAS submodel elements in published submodel templates.

IDTA Number	Short Name	ARel	Blob	Capability	Entity	File	MLP	Operation	Property	Range	Ref	Rel	SMC
02001-1-0	Module Type Package					✓	✓					✓	✓
02002-1-0	Contact Information						✓		✓				✓
02003-1-2	Technical Data					✓	✓		✓				✓
02004-1-2	Handover Documentation				✓	✓	✓		✓		✓		✓
02005-1-0	Simulation Models					✓	✓		✓				✓
02006-2-0	Digital Nameplate					✓	✓		✓		✓		✓
02008-1-1	Time Series Data		✓			✓	✓	✓	✓	✓			✓
02013-1-0	Reliability								✓				✓
02014-1-0	Functional Safety								✓				✓
	Test Model		✓		✓	✓	✓	✓	✓	✓	✓	✓	✓

Abbreviations: ARel, AnnotatedRelationshipElement; MLP, MultiLanguageProperty; Ref, ReferenceElement; Rel, RelationshipElement; SMC, SubmodelElementCollection.

**Table 3 sensors-23-05229-t003:** Protocol-specific APIs of the test assets.

	HTTP	MQTT	OPC UA
Read	HTTP GET /value	-	NodeId: nsu=com:example:test;s=Node1
	Result: { “value”: ? }		
Write	HTTP PUT /value	Topic: data/value	NodeId: nsu=com:example:test;s=Node1
	Payload: { “value”: ? }	Payload: { “value”: ? }	
Execute	HTTP POST /add	-	NodeId: nsu=com:example:test;s=Operation1
	Payload:		Input: Input1, Input2
	{		Output: Result
	“data”: {		
	“input1”: ?,		
	“input2”: ?,		
	}		
	}		
	Result: { “result”: ? }		
Subscribe	-	Topic: data/value	NodeId: nsu=com:example:test;s=Node1
		Payload: { “value”: ? }	

**Table 4 sensors-23-05229-t004:** Evaluation results of the comparison of open-source AAS Type 2 implementations.

	AASX Server	Eclipse BaSyx	FA^3^ST Service	NOVAAS
General Criteria				
Main Contributors	Phoenix Contact Zurich University of Applied Sciences Fraunhofer IOSB-INA Festo SE & Co. KG	Fraunhofer IESE objective partner AG	Fraunhofer IOSB	NOVA School of Science and Technology
License	Apache v2.0	MIT	Apache v2.0	EUPL v1.2
Programming Language	C#	Java	Java	Node-RED (JavaScript)
Usage				
CLI	✓	✓	✓	-
Docker	✓	✓	✓	✓
Embedded	-	✓	✓	-
AAS-Specific Criteria				
Metamodel Version	v2/v3 *	v2	v3	v2
Model Format				
AASX	✓	✓	✓	(✓)
JSON	(✓)	✓	✓	(✓)
XML	(✓)	✓	✓	-
RDF	-	-	✓	-
API Interfaces				
AAS Repository	-/✓ *	✓	✓	(✓)
AAS	(✓)/✓ *	✓	✓	(✓)
SM Repository	-/✓ *	-	✓	✓
SM	(✓)/✓ *	(✓)	✓	✓
CD Repository	-/✓ *	-	✓	-
AAS Serialization	-/✓ *	-	✓	-
AAS Basic Discovery	-/✓ *	-	✓	-
AASX File Server	(✓)/✓ *	-	-	-
Descriptor	-	-	-	-
API Protocols				
HTTP	✓	✓	✓	✓
OPC UA	✓	-	✓	-
Implementation-specific Criteria				
Asset Synchronization				
HTTP				
Read	-	✓ ^◊^	✓	(✓) ^†^
Write	-	(✓)	✓	-
Execute	-	✓ ^†,◊^	✓	-
OPC UA				
Read	✓	(✓) ^†,Δ^	✓	(✓) ^†^
Write	-	(✓) ^†^	✓	-
Execute	-	(✓) ^†^	✓	-
Subscribe	(✓)	(✓) ^†,Δ^	✓	-
MQTT				
Write	-	(✓) ^†^	✓	-
Subscribe	-	(✓) ^†,Δ^	✓	-
Data Persistence				
In-Memory	✓	✓	✓	✓
File	-	(✓) ^†^	✓	-
Database	-	✓	-	-
Security	(✓)	✓	-	(✓)
Other Features	Publish value changes via MQTT	Publish any changes via MQTTCustom extensions via feature-decorators	Parallel synchronized endpointsOpen Architecture	No/Low-code approachPublish value changes via MQTT
Ecosystem Criteria				
Client Library	(✓)	✓	-	-
Model Editor	✓	-	-	-
Model Validation	-	(✓)	✓	-
Registry	(✓)	✓	-	-
Visualization	✓	✓	-	✓
Other	Integrated with AASX Package ExplorerXLS Parser	DataBridgeUI Plug-in MechanismKeyCloack Integration	Eclipse Dataspace Connectorv2 Model Converter	DashboardKeyCloak Integration

* only in undocumented API; ^†^ via custom coding; ^◊^ via delegation; ^Δ^ via DataBridge. Abbreviations: AAS, Asset Administration Shell; AASX, Asset Administration Shell Exchange Format; CD, Concept Description; CLI, Command-Line Interface; SM, Submodel; SMT, Submodel Template.

**Table 5 sensors-23-05229-t005:** Results of AAS API compatibility test.

	AASX Server	Eclipse BaSyx	FA^3^ST Service	NOVAAS
AAS Repository Interface				
GetAllAssetAdministrationShells	✓	✓	✓	✓
GetAssetAdministrationShellsById	✓	✓	✓	✓
PostAssetAdministrationShell	-/✓ *	(✓)	✓	-
PutAssetAdministrationShellById	-/✓ *	✓	✓	-
DeleteAssetAdministrationShellById	✓	✓	✓	-
AAS Interface				
GetAssetAdministrationShell	✓	✓	✓	✓
GetAllSubmodelReferences	-/✓ *	✓	✓	✓
PostSubmodelReference	-/✓ *	✓	✓	✓
DeleteSubmodelReference	-/✓ *	✓	✓	✓
GetAssetInformation	-/✓ *	-	✓	✓
Submodel Interface				
GetSubmodel	✓	✓	✓	✓
level = core	✓	-	✓	✓
GetAllSubmodelElements	-/✓ *	✓	✓	-
GetSubmodelElementByPath	✓	✓	✓	✓
content = metadata	-	-	-	✓
content = value	✓	✓	✓	✓
extent = withBlobValue	-/✓ *	-	✓	✓
PostSubmodelElement	✓	✓	✓	✓
PostSubmodelElementByPath	✓	✓	✓	✓
PutSubmodelElementByPath	✓	✓	✓	✓
DeleteSubmodelElementByPath	✓	✓	✓	✓

* only in undocumented API.

## Data Availability

No new data were created or analyzed in this study. Data sharing is not applicable to this article.

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
