# Peer review of "Open-Source Implementations of the Reactive Asset Administration Shell: A Survey"

_sensors, 2023, doi:10.3390/s23115229_

Round 1
Reviewer 1 Report
Congratulations for the authors. For me very good article presenting a lot of data & conclusions and work that had to be done to obtain the 2 mentioned above. I think article can be accepted in a present form... I don't have any ideas/suggestions how to improve it, I didn't notice any part missing, article has a good structure, everything is explained plain & simple (although topic is a "hard one"). Job well done!
Author Response
Thank you for your positive and encouraging feedback!
Reviewer 2 Report
This is a very good paper. It would be highly useful to those who are evaluating approaches to the AAS implementation. The authors use good criteria, and the addition of synchronization with the physical object is a useful addition.
This is a very good paper. It would be highly useful to those who are evaluating approaches to the AAS implementation. The authors use good criteria, and the addition of synchronization with the physical object is a useful addition.
The results need to be assumed as correct, since the only way to check that would be to do the work that they did. That obviously isn’t practical.
This brings me to my one concern and that is bias. The authors are evaluating one of their own products. While I have no doubt in their professionalism in doing the work that they said, this probably should be highlighted. They do address this in a fashion at the end of the paper, but it should be noted and highlighted much earlier than that.
This probably should be highlighted. They do address this in a fashion at the end of the paper, but it should be noted much earlier than that.
That said, I think this is an excellent paper.
Line 381 has a grammatical error. It should say “that implements“. The paper should be read through one more time in case there are other minor grammatical errors like this.
Author Response
Thank you for your valuable feedback. We addressed your feedback regarding our bias and now additionally mention our bias/conflict of interest early in the paper. We also improved the grammar as you suggested.
Reviewer 3 Report
- In introduction, the main contribution and originality should be explained in more detail.
other abbreviations must be mentioned in the list.
- The study proposed in this article should be justified by other references.
Author Response
Thank you for your valuable feedback. Accordign to your recommendations, we added a paragraph in the introduction highlighting the main contributions of the paper.
We also made sure that all abbreviations are mentioned in the list of abbreviations.
Unfortunately, your last feedback point "The study proposed in this article should be justified by other references" is not clear to us. What exactly should be justified? The necessity for this work? This we have shown by highlighting the relevance of the topic and the shortcomings of existing comparisons.
Reviewer 4 Report
The topic of this article is interesting. The objective is presented clearly. However, the authors may revise this article to enhance its quality by considering the following concerns:
1) The abstract content is not clear enough. It can be written according to the logical structure of the paper.
2) I would suggest including a background introduction with contexts related to the topics so that the readers have a good understanding of the overall concept.
3) The authors should provide the analysis of related works in a much more thoroughly way. When citing existing literature, please do not just say what others done something, but should also discuss what's the relevance of others' work to this paper.
4)The list of references has some style problem. The style of some references is inconsistence with the format required.
Minor editing of English language required
Author Response
Thank you for your valuable feedback. Please find our comments on your feedback below Comment: The abstract content is not clear enough. It can be written according to the logical structure of the paper. Response: The abstract is actually following the logical structure of the paper quite in detail. It also includes all aspects typically required for an abstract to contain such as background information / motivation, brief description of methods, principal results, conclusions/interpretations as explained on the MDPI website (https://www.mdpi.com/authors/layout#_bookmark5) Comment: I would suggest including a background introduction with contexts related to the topics so that the readers have a good understanding of the overall concept. Response: We included an additional paragraph and two figures with additional background information on the Asset Administration Shell. For more details on e.g. the underlying specification the references to the original documents are provided. Comment: The authors should provide the analysis of related works in a much more thoroughly way. When citing existing literature, please do not just say what others done something, but should also discuss what's the relevance of others' work to this paper. Response: Assumingly this is refering to other papers in the related works section that are also comparing different implementations of the AAS specification. For those we discuss the relevance by summarizing the content, analyzing them in short and relating them to our work, e.g. by highlighting the drawbacks of existing papers and how these issues are addressed in our research. From our perspective, this is capturing the relevance of existing literature in the context of our work. Comment: The list of references has some style problem. The style of some references is inconsistence with the format required. Response: We double checked the references with the styleguide provided from the Sensors Journal (https://www.mdpi.com/journal/sensors/instructions) as well as the general MDPI one (https://www.mdpi.com/authors/layout#_bookmark93) and could not find any format inconcistencies. If you could provide more detailed feedback on what exactly is wrong with references we are happy to address this issue. Comment: Minor editing of English language required Response: We double checked grammar and made a few updates.